# Consistent Comparison of Symptom-based Methods for COVID-19 Infection Detection (Extended Abstract)

Jesús Rufino[1], Juan Marcos Ramírez[1], Jose Aguilar[1], Carlos Baquero[2], Jaya Champati[1], Davide Frey[3], Rosa Elvira Lillo-Rodríguez[4], Antonio Fernández Anta[1]

[1]IMDEA Networks Institute, Madrid, Spain, [2]Universidade do Minho, Braga, Portugal
[3]INRIA, Rennes, France, [4] Universidad Carlos III, Madrid, Spain

## ABSTRACT

During the global pandemic crisis, several COVID-19 diagnosis methods based on survey information have been proposed with the purpose of providing medical staff with quick detection tools that allow them to efficiently plan the limited healthcare resources. In general, these methods have been developed to detect COVID-19-positive cases from a particular combination of self-reported symptoms. In addition, these methods have been evaluated using datasets extracted from different studies with different characteristics. On the other hand, the University of Maryland, in partnership with Facebook, launched the Global COVID-19 Trends and Impact Survey (UMD-CTIS), the largest health surveillance tool to date that has collected information from 114 countries/territories from April 2020 to June 2022. This survey collected information on various individual features including gender, age groups, self-reported symptoms, isolation measures, and mental health status, among others. In this paper, we compare the performance of different COVID-19 diagnosis methods using the information collected by UMD-CTIS, for the years 2020 and 2021, in six countries: Brazil, Canada, Israel, Japan, Turkey, and South Africa. The evaluation of these methods with homogeneous data across countries and years provides a solid and consistent comparison among them.

## KEYWORDS

COVID-19 diagnosis, F1-score, light gradient boosting machine, logistic regression, rule-based methods.

## 1 INTRODUCTION

In December 2019, the *coronavirus disease 2019* (COVID-19) emerged in China caused by the *severe acute respiratory syndrome coronavirus 2* (SARS-CoV-2) [17]. Within a few months, this disease led to a global pandemic crisis that has challenged national healthcare systems [6]. More precisely, by June 2023, the cumulative number of confirmed cases worldwide exceeded 688 million, and officially over 6,800,000 people have died from COVID-19; https://www.worldometers.info/coronavirus/. In this context, the planning of the healthcare resources (e.g., the estimation of the number of hospital beds or intensive care units needed for COVID-19 patients) has been determined by the availability of quick and efficient instruments for the diagnosis of active cases.

The *reverse transcriptase-polymerase chain reaction* (RT-PCR) test has been considered the standard tool to detect infected people [5]. However, real-time disease monitoring based on the RT-PCR test demands material and human resources that are not always available. To overcome these limitations, various diagnosis methods based on survey information have been proposed that combine multiple individual features (age, gender, symptoms, demographic data, etc.) to

characterize COVID-19-infected people [1–4, 9–12, 14–16, 18, 19]. Specifically, most of these methods propose simple rules or build machine learning models that evaluate a set of individual attributes to determine a COVID-19-positive case. However, a consistent comparison framework that evaluates the performance yielded by the different methods is missing since the generated models and the corresponding conclusions are assessed using different datasets that are heterogeneous in size and type.

On the other hand, in April 2020, the University of Maryland Global COVID-19 Trends and Impact Survey (UMD-CTIS), in partnership with Facebook, launched the largest global health surveillance platform to date [8]. More precisely, this project stored the responses provided by a subset of Facebook invited users about different topics related to the COVID-19 pandemic such as the presence of symptoms, RT-PCR outcomes, and vaccination acceptance, among others. This data collection instrument was available in 56 languages and it recorded tens of millions of responses from 114 countries or territories worldwide.

In this paper, we conduct a consistent comparison of different methods that detect COVID-19-positive cases from a combination of features collected from surveys. To this end, we take into account the information included in the UMD-CTIS records extracted from six countries: Brazil, Canada, Israel, Japan, Turkey, and South Africa. For each country, the models are trained using a randomly selected subset of tested individuals who reported at least one symptom. Furthermore, we compare the performance for two years: 2020 and 2021, which represent two different periods of the pandemic without and with vaccination, respectively. We compare the detection methods using four performance metrics: $F_1$-score, sensitivity, specificity, and precision (only $F_1$-score is presented in this extended abstract). Overall, the detection methods exhibiting the best performances across different groups and metrics are **Mika** [10] ($F_1$-score: 59.33%), **Astley** [3] ($F_1$-score: 59.22%), **Smith** [16] ($F_1$-score: 59.22%), **Bhattacharya** [4] ($F_1$-score: 58.69%), **Roland** [12] ($F_1$-score: 58.20%), **Shoer** [15] ($F_1$-score: 58.15%), **Menni_1** [9] ($F_1$-score: 57.03%), and **Menni_2** [9] ($F_1$-score: 56.94%).

## 2 MATERIALS AND METHODS

### 2.1 UMD-CTIS Survey

We perform a consistent comparative study of various COVID-19 active case detection methods from data provided by the UMD-CTIS survey. More precisely, since April 23, 2020, Facebook worldwide users were invited to participate in the UMD-CTIS survey. Users who accepted the invitation were moved to a web survey platform, where potential participants must report age > 18 and consent of data use before responding to the survey. The survey instrument consists of a web-based questionnaire collecting information on

gender, age groups, symptoms, COVID testing, isolation, and vaccination, among others. Furthermore, the survey instrument was continuously updated to aggregate new items. Finally, UMD organized and stored daily microdata that were further processed to develop our comparative study.

## 2.2 Comparative study design

In this work, we compare the performance of various COVID-19 detection methods using the information provided by UMD-CTIS data extracted from six countries: Brazil, Canada, Israel, Japan, Turkey, and South Africa. These countries are selected based on geographical diversity and the large amount of available data. In addition, this comparative study is performed for two non-overlapped periods: (2020) from April 23 to December 31, 2020, and (2021) from January 1 to December 31, 2021. Notice that the end of 2020 matches the start of the first COVID-19 vaccination campaigns. Therefore, we can compare the performance of the detection methods without and with information on vaccination. Table 1 summarizes the characteristics of the study population for the various countries and for the two periods under test.

For every country and period, we build a dataset by picking the answers reporting lab test results in the last 14 days (the survey does not collect the test type) and at least one potential COVID-19 symptom, i.e., this comparative study selects the tested and symptomatic cases. We select symptomatic cases because feature-based predictive methods typically aim at finding the combination of symptoms that detect infected people. In addition, we choose the tested individuals with the aim of obtaining the ground truth sample set that allows us to evaluate the performance of the different methods quantitatively. Since questionnaires contain categorical data, we apply binary encoding (dummy coding) to each response. This leads to datasets with 201 features (attributes, columns, or variables) for 2020, and the datasets have between 431 and 452 columns for 2021 depending on the selected country. For each dataset, this study evaluates the performance of the various COVID-19 active case detection methods. To this end, our study divided every dataset into 100 partitions. For each trial, 80% of the dataset rows (questionnaires or samples) were randomly selected as training samples, and the remaining 20% were used to test the various methods.

## 2.3 Detection methods under comparison

In this work, we compare the performance of various COVID-19 diagnosis methods belonging to three categories:

(1) Rule-based methods: CDC [1], WHO [18], Akimbami [2], Solomon [14], Perez [11].
(2) Logistic regression techniques: Menni [9], Roland [12], Smith [16], Shoer [15], Bhattacharya [4], Mika [10].
(3) Tree-based machine-learning models: Zoabi [19], Astley [3].

In this work, we have implemented two versions of the Menni method and two versions of the Zoabi method. Note that UMD-CTIS data did not register whether the respondent skipped meals. Therefore, we modified the Menni method by fixing the *skipped meals* variable to zero (**Menni_1**). Furthermore, we followed the procedure reported in [9] to build the logistic regression model from individual features available in our dataset (**Menni_2**). In other words, we built a regression model that considers the features:

age, gender, loss of smell and taste, cough, and fatigue. In the case of the Zoabi method, notice that UMD-CTIS data ranges of ages do not have a boundary at 60. The boundary is either at 55 or 65. We have created two different models, one for ages greater than 55 years (**Zoabi_55**) and the other for ages greater than 65 years (**Zoabi_65**). Further information regarding the methods under test can be found in the corresponding references and in the full version of the article [13].

## 2.4 Benchmarking detection methods

First, we use the $F_1$-score to quantitatively assess the performance of the various detection methods. To this end, our procedure firstly obtains the predictions over the test set for each trial. From the predicted estimates and the ground truth data, the procedure identifies the number of true positives TP, false positives FP, true negatives TN, and false negatives FN. Then, the $F_1$-score is obtained as follows:

$$F_1 = \frac{2TP}{2TP + FP + FN}. \tag{1}$$

Tables 2 and 3 display the ensemble average and the CI of the $F_1$-score for the five countries and for 2020 and 2021, respectively. Specifically, each value in these tables is obtained by averaging 100 realizations of the corresponding experiment. Tables with the sensitivity, specificity, and precision values obtained are included in the full version of the article [13].

## 3 RESULTS

As can be seen in Table 1, 83, 238 respondents from Brazil reported a test outcome and at least one symptom in 2020. In this cohort, 44, 963 participants reported a positive test result, and 38, 275 respondents had a negative test outcome. Table 1 also includes the test positive rate (TPR) where TPR $= (100 \times \text{positive})/(\text{Tested symptomatic})$. For example, the TPR for Brazil 2020 is 54.02%. On the other hand, for Brazil 2021, the dataset was extracted from 262, 683 participants who reported at least one symptom and the outcome of a test done in the last 14 days. In this case, 106, 471 respondents reported a positive test result, and 156, 212 questionnaires informed a negative test outcome with a TPR of 40.53%. In summary, the number of tested symptomatic, the number of positive cases, and the number of negative results for the remaining countries in 2020 and 2021 are displayed in Table 1. Additionally, Table 1 shows information about other individual features such as gender and age groups.

Table 2 shows the ensemble averages with the corresponding 95% confidence intervals (CI) of the $F_1$ score yielded by the various detection methods for the different countries and for 2020. In particular, the methods the best $F_1$ scores for each country are: Brazil (**Astley**: 73.72%), Canada (**Menni_1**: 54.33%), Israel (**Bhattacharya**: 62.78%), Japan (**Menni_1**: 46.33%), Turkey (**Bhattacharya**: 67.67%), and South Africa (**Roland**: 67.32%). The $F_1$ score in % and the CIs obtained for 2021 are displayed in Table 3. For 2021, the best $F_1$ scores are: Brazil (**Menni_2**: 66.54%), Canada (**Smith**: 50.28%), Israel (**Bhattacharya**: 58.76%), Japan (**Mika**: 52.41%), Turkey (**Bhattacharya**: 64.61%), and South Africa (**Menni_2**: 66.50%). As observed in Tables 2 and 3, none of the methods achieved an $F_1$ score of 74% or above, indicating that no model is very good. According to Table 1, Brazil, Turkey, and South Africa exhibit TPR values at least twofold higher than those obtained from Canada, Israel, and Japan.

**Table 1: Characteristics of the study population for the various countries and for two non-overlapped periods (2020 and 2021).**

| | Characteristic | Brazil | | Canada | | Israel | | Japan | | Turkey | | South Africa | |
|---|---|---|---|---|---|---|---|---|---|---|---|---|---|
| | | 2020 | 2021 | 2020 | 2021 | 2020 | 2021 | 2020 | 2021 | 2020 | 2021 | 2020 | 2021 |
| 1. | Tested symptomatic, N | 83238 | 262683 | 8927 | 33997 | 5944 | 19063 | 4698 | 41010 | 15952 | 28896 | 7883 | 23038 |
| 2. | Test outcome | | | | | | | | | | | | |
| | (a) Positive, N | 44963 | 106471 | 838 | 3433 | 1238 | 2869 | 532 | 4011 | 6167 | 9228 | 2866 | 8459 |
| | (b) Negative, N | 38275 | 156212 | 8089 | 30564 | 4706 | 16194 | 4166 | 36999 | 9785 | 19668 | 5017 | 14579 |
| | (c) TPR, % | 54.02 | 40.53 | 9.39 | 10.10 | 20.83 | 15.05 | 11.32 | 9.78 | 38.66 | 31.94 | 36.35 | 36.71 |
| 3. | Gender | | | | | | | | | | | | |
| | (a) Female, N | 45357 | 130235 | 5438 | 19472 | 2941 | 9290 | 1679 | 14283 | 3939 | 7185 | 3923 | 11291 |
| | (b) Male, N | 24928 | 76689 | 2315 | 9824 | 2199 | 6746 | 2388 | 20791 | 8920 | 15292 | 2525 | 6730 |
| 4. | Age groups | | | | | | | | | | | | |
| | (a) 18-24, N | 8270 | 27474 | 1136 | 3248 | 583 | 1498 | 179 | 871 | 1716 | 2267 | 739 | 1580 |
| | (b) 25-34, N | 19596 | 56227 | 2337 | 7172 | 1144 | 3069 | 577 | 3797 | 4375 | 5756 | 2252 | 4889 |
| | (c) 35-44, N | 21061 | 57452 | 1750 | 6688 | 1041 | 3333 | 997 | 7527 | 4043 | 7110 | 1801 | 4721 |
| | (d) 45-54, N | 13776 | 39122 | 1210 | 5215 | 933 | 3115 | 1216 | 10413 | 2071 | 4594 | 1141 | 3878 |
| | (e) 55-64, N | 6968 | 22190 | 954 | 4478 | 880 | 2634 | 828 | 8724 | 862 | 2400 | 491 | 2124 |
| | (f) 65-74, N | 140 | 6016 | 308 | 2421 | 510 | 1957 | 479 | 3529 | 158 | 719 | 1667 | 799 |
| | (g) 75+, N | 233 | 1025 | 126 | 825 | 143 | 627 | 66 | 846 | 21 | 134 | 27 | 230 |

**Table 2: $F_1$ score and its $95\%$ confidence interval for the selected countries for 2020, in $\%$.**

| Method | Brazil | Canada | Israel | Japan | Turkey | South Africa |
|---|---|---|---|---|---|---|
| Menni_1 | 65.56 (65.48 - 65.64) | 54.33 (53.66 - 54.99) | 59.76 (59.16 - 60.36) | 46.33 (45.33 - 47.33) | 63.93 (63.68 - 64.17) | 61.39 (61.07 - 61.70) |
| Menni_2 | 71.13 (71.01 - 71.24) | 49.33(48.77 - 49.88) | 57.50 (57.04 - 57.97) | 39.91 (39.27 - 40.54) | 67.41 (67.21 - 67.60) | 66.36 (66.10 - 66.62) |
| Roland | 69.38 (69.30 - 69.46) | 51.44 (50.86 - 52.02) | 61.93 (61.46 - 62.41) | 40.68 (39.98 - 41.39) | 67.06 (66.87 - 67.26) | 67.32 (67.05 - 67.58) |
| Smith | 71.11 (71.05 - 71.18) | 53.43 (52.85 - 54.01) | 62.47 (61.98 - 62.97) | 45.12 (44.42 - 45.82) | 67.30 (67.11 - 67.49) | 62.06 (61.80 - 62.32) |
| Zoabi_55 | 70.71 (70.65 - 70.77) | 32.96 (32.37 - 33.54) | 47.76 (47.32 - 48.20) | 29.95 (29.29 - 30.60) | 57.86 (57.69 - 58.03) | 59.05 (58.80 - 59.31) |
| Zoabi_65 | 70.73 (70.67 - 70.79) | 32.86 (32.28 - 33.44) | 47.79 (47.36 - 48.23) | 29.91 (29.27 - 30.55) | 57.72 (57.55 - 57.88) | 59.00 (58.74 - 59.25) |
| CDC | 73.42 (73.36 - 73.48) | 23.43 (23.14 - 23.72) | 45.84 (45.46 - 46.21) | 27.38 (27.00 - 27.75) | 62.60 (62.42 - 62.78) | 62.13 (61.88 - 62.39) |
| Shoer | 70.45 (70.39 - 70.52) | 50.95 (50.37 - 51.54) | 62.41 (61.93 - 62.89) | 44.57 (43.86 - 45.28) | 67.49 (67.30 - 67.69) | 66.76 (66.52 - 67.00) |
| Bhattacharya | 69.77 (69.70 - 69.83) | 51.90 (51.31 - 52.50) | 62.78 (62.30 - 63.26) | 39.41 (38.84 - 39.97) | 67.67 (67.48 - 67.87) | 66.81 (66.52 - 67.10) |
| WHO | 23.92 (23.83 - 24.01) | 24.08 (23.45 - 24.70) | 24.69 (24.15 - 25.24) | 27.29 (26.52 - 28.06) | 25.14 (24.90 - 25.38) | 30.97 (30.59 - 31.35) |
| Perez | 59.47 (59.39 - 59.55) | 45.20 (44.56 - 45.83) | 52.27 (51.71 - 52.82) | 32.93 (32.23 - 33.64) | 58.12 (57.89 - 58.35) | 61.00 (60.70 - 61.30) |
| Mika | 69.43 (69.37 - 69.49) | 51.43 (50.86 - 52.01) | 62.16 (61.68 - 62.63) | 45.29 (44.65 - 45.94) | 67.08 (66.89 - 67.28) | 66.40 (66.13 - 66.68) |
| Akinbami_1 | 12.85 (12.77 - 12.94) | 11.33 (10.72 - 11.93) | 10.22 (9.82 - 10.62) | 13.38 (12.58 - 14.18) | 11.48 (11.26 - 11.70) | 17.70 (17.34 - 18.07) |
| Akinbami_2 | 14.69 (14.60 - 14.78) | 9.41 (8.89 - 9.92) | 9.59 (9.16 - 10.01) | 13.16 (12.35 - 13.98) | 10.81 (10.60 - 11.03) | 17.14 (16.80 - 17.49) |
| Akinbami_3 | 27.84 (27.73 - 27.94) | 20.23 (19.66 - 20.81) | 21.67 (21.14- 22.19) | 18.98 (18.22 - 19.73) | 26.31 (26.05 - 26.56) | 28.93 (28.57 - 29.29) |
| Salomon | 30.97 (30.87 - 31.07) | 25.52 (24.84 - 26.20) | 27.12 (26.58 - 27.66) | 30.64 (29.93 - 31.35) | 28.36 (28.10 - 28.61) | 39.35 (38.98 - 39.72) |
| Astley | 73.72 (73.65 - 73.78) | 48.29 (47.58 - 49.00) | 62.47 (61.98 - 62.97) | 44.13 (43.32 - 44.93) | 67.45 (67.24 - 67.65) | 66.85 (66.61 - 67.09) |

Since the $F_1$ score is highly affected by imbalanced classes [7], we computed the averages of the $F_1$ score yielded by the detection methods for three groups: the broad set of the six countries, the set of countries with high TPR (Brazil, Turkey, and South Africa) and low TPR (Canada, Israel, and Japan) for 2020, 2021, and the entire interval 2020-2021 (Table 4). For 2020, when there was no vaccination yet, the most efficient method was Astley (Average: 60.49%). In the Astley method, the most relevant are cough, stuffy or runny nose, aches or muscle pain, headache, sore throat, and fever. In 2021, when vaccination began, Mika was the most effective method (Average: 58.35%). In the Mika method, fever, cough, loss of taste and smell, and gastrointestinal problems are considered for COVID-19 detection. In the full article [13], we compared the various detection methods in terms of sensitivity, specificity, and precision.

## 4 CONCLUSIONS

In this work, we conduct a comparison of various COVID-19 diagnosis methods based on survey information using datasets extracted from the global UMD-CTIS survey. More precisely, we compare the different methods for six countries and two periods (with and without vaccines) using the $F_1$ score as a performance metric. From these results, we highlight the techniques showing the best $F_1$ score. It is important to mention that, as can be seen in Tables 2 and 3, none of the methods achieve an $F_1$ score above 75% indicating that no model has a superior performance.

Additional results and a more extended discussion can be found in the full version of the article [13].

## 5 ETHICAL DECLARATION

The Ethics Board (IRB) of IMDEA Networks Institute gave ethical approval for this work on 2021/07/05. IMDEA Networks has signed Data Use Agreements with Facebook and the University of Maryland (UMD) to access their data, specifically, UMD project 1587016-3 entitled C-SPEC: Symptom Survey: COVID-19 entitled ILI Community-Surveillance Study. The data used in this study was collected by the University of Maryland through The University of Maryland Social Data Science Center Global COVID-19 Trends and Impact Survey in partnership with Facebook. Informed consent has been obtained from all participants in this survey by this institution. All the methods in this study have been carried out in accordance with relevant ethics and privacy guidelines and regulations.

## 6 AVAILABILITY OF DATA AND MATERIALS

The data presented in this paper (in aggregated form) and the programs used to process it will be openly accessible at https://github.com/GCGImdea/coronasurveys/. The microdata of the CTIS survey from which the aggregated data was obtained cannot be shared, as per the Data Use Agreements signed with Facebook and the University of Maryland (UMD).

## 7 FUNDING/SUPPORT

This work was partially supported by grants COMODIN-CM and PredCov-CM, funded by Comunidad de Madrid and the European Union through the European Regional Development Fund (ERDF), and grants TED2021-131264B-I00 (SocialProbing) and PID2019-104901RB-I00, funded by Ministry of Science and Innovation - State

**Table 3: $F_1$ score and its 95% confidence interval for the selected countries for 2021, in %**

| Method | Brazil | Canada | Israel | Japan | Turkey | South Africa |
|---|---|---|---|---|---|---|
| Menni_1 | 59.24 (59.18 - 59.31) | 49.38 (49.02- 49.74) | 57.31 (56.96 - 57.65) | 49.24 (49.16 - 49.83) | 59.65 (59.44 - 59.87) | 58.28 (58.06 - 58.50) |
| Menni_2 | 66.54 (66.49 - 66.59) | 39.82 (39.59 - 40.05) | 53.46 (53.21 - 53.70) | 42.60 (42.37 - 42.84) | 62.71 (62.56 - 62.85) | 66.50 (66.33 - 66.68) |
| Roland | 65.76 (65.71 - 65.82) | 46.28 (46.03 - 46.53) | 57.16 (56.86 - 57.46) | 42.82 (42.62 - 43.03) | 64.13 (63.96 - 64.31) | 64.41 (64.23 - 64.59) |
| Smith | 63.37 (63.32 - 63.42) | 50.28 (49.99 - 50.57) | 58.00 (57.68 - 58.33) | 51.48 (51.23 -51.74) | 64.38 (64.21 - 64.55) | 61.62 (61.45 - 61.80) |
| Zoabi_55 | 59.83 (59.79 - 59.88) | 37.31 (37.01 - 37.60) | 39.63 (39.28 - 39.98) | 33.71 (33.45 - 33.98) | 52.14 (51.88 - 52.40) | 59.62 (59.47 - 59.77) |
| Zoabi_65 | 59.78 (59.74 - 59.83) | 37.10 (36.81 - 37.39) | 39.64 (39.29 - 39.99) | 33.36 (33.11 - 33.62) | 52.06 (51.80 - 52.31) | 59.54 (59.38 - 59.69) |
| CDC | 63.22 (63.17 - 63.26) | 27.41 (27.28 - 27.55) | 38.78 (38.59 - 38.97) | 28.54 (28.40 - 28.68) | 55.96 (55.81 - 56.11) | 61.25 (61.10 - 61.39) |
| Shoer | 65.81 (65.76 - 65.87) | 41.10 (40.84 - 41.36) | 53.67 (53.37 - 53.97) | 45.42 (45.07 - 45.78) | 64.18 (64.01 - 64.35) | 64.97 (64.80 - 65.15) |
| Bhattacharya | 64.16 (64.11 - 64.22) | 49.22 (48.96 - 49.49) | 58.76 (58.48 - 59.03) | 45.82 (45.59 - 46.05) | 64.61 (64.44 - 64.78) | 63.40 (63.22 - 63.59) |
| WHO | 23.62 (23.56 - 23.68) | 26.01 (25.66 - 26.35) | 27.92 (27.59 - 28.24) | 34.05 (33.74 - 34.37) | 27.72 (27.49 - 27.94) | 32.78 (32.58 - 32.98) |
| Perez | 54.85 (54.79 - 54.90) | 44.70 (44.40 - 45.00) | 51.27 (50.93 - 51.61) | 39.72 (39.45 - 40.00) | 56.03 (55.86 - 56.21) | 59.17 (58.98 - 59.35) |
| Mika | 65.33 (65.28 - 65.38) | 46.76 (46.40 - 47.12) | 57.50 (57.22 - 57.79) | 52.41 (51.73 - 53.09) | 64.13 (63.96 - 64.31) | 63.98 (63.81 - 64.15) |
| Akinbami_1 | 12.02 (11.96 - 12.07) | 11.43 (11.17 - 11.70) | 10.60 (10.33 - 10.88) | 11.11 (10.82 - 11.39) | 13.86 (13.69 - 14.03) | 15.86 (15.66 - 16.06) |
| Akinbami_2 | 12.02 (12.05 - 12.16) | 8.03 (7.79 - 8.27) | 11.48 (11.20 - 11.75) | 9.10 (8.83 - 9.31) | 11.80 (11.64 - 11.96) | 13.61 (13.44 - 13.79) |
| Akinbami_3 | 26.59 (26.00 - 26.11) | 20.96 (20.64 - 21.27) | 21.96 21.62 - 22.30) | 19.90 (19.63 - 20.17) | 26.35 (26.12 - 26.58) | 28.08 (27.85 - 28.31) |
| Salomon | 30.15 (30.11 - 30.24) | 28.06 (27.70 - 28.43) | 30.72 (30.39 - 31.05) | 37.27 (36.97 - 37.57) | 31.31 (31.09 - 31.53) | 38.03 (37.83 - 38.23) |
| Astley | 65.95 (65.90 - 66.01) | 45.07 (44.74 - 45.40) | 58.62 (58.29 - 58.94) | 50.39 (50.08 - 50.70) | 63.67 (63.50 - 63.85) | 64.06 (63.88 - 64.24) |

**Table 4: Average $F_1$ score (in %) for three country groups: the overall six countries (overall), the countries with high TPR (High TPR: Brazil, Turkey, and South Africa), and the countries with low TPR (Low TPR: Canada, Israel, and Japan) for 2020, 2021, 2020-2021.**

| Method | 2020 | | | 2021 | | | 2020-2021 | | |
|---|---|---|---|---|---|---|---|---|---|
| | Overall | Low TPR | TPR TPR | Overall | Low TPR | High TPR | Overall | Low TPR | High TPR |
| Menni_1 | 58.55 | 53.47 | 63.63 | 55.52 | 51.98 | 59.06 | 57.03 | 52.73 | 61.34 |
| Menni_2 | 58.61 | 48.91 | 68.30 | 55.27 | 45.29 | 65.25 | 56.94 | 47.10 | 66.78 |
| Roland | 59.64 | 51.35 | 67.92 | 56.76 | 48.75 | 64.77 | 58.20 | 50.05 | 66.34 |
| Smith | 60.25 | 53.67 | 66.82 | 58.19 | 53.25 | 63.12 | 59.22 | 53.46 | 64.97 |
| Zoabi_55 | 49.72 | 36.89 | 62.54 | 47.04 | 36.88 | 57.20 | 48.38 | 36.89 | 59.87 |
| Zoabi_65 | 49.67 | 36.85 | 62.48 | 46.91 | 36.70 | 57.13 | 48.29 | 36.78 | 59.81 |
| CDC | 49.13 | 32.22 | 66.05 | 45.86 | 31.58 | 60.14 | 47.50 | 31.90 | 63.10 |
| Shoer | 60.44 | 52.64 | 68.23 | 55.86 | 46.73 | 64.99 | 58.15 | 49.69 | 66.61 |
| Bhattacharya | 59.72 | 51.36 | 68.08 | 57.66 | 51.27 | 64.06 | 58.69 | 51.32 | 66.07 |
| WHO | 26.02 | 25.35 | 26.68 | 28.68 | 29.33 | 28.04 | 27.35 | 27.34 | 27.36 |
| Perez | 51.50 | 43.47 | 59.53 | 50.96 | 45.23 | 56.68 | 51.23 | 44.35 | 58.11 |
| Mika | 60.30 | 52.96 | 67.64 | 58.35 | 52.22 | 64.48 | 59.33 | 52.59 | 66.06 |
| Akinbami_1 | 12.83 | 11.64 | 14.01 | 12.48 | 11.05 | 13.91 | 12.65 | 11.35 | 13.96 |
| Akinbami_2 | 12.47 | 10.72 | 14.21 | 11.02 | 9.54 | 12.51 | 11.75 | 10.13 | 13.36 |
| Akinbami_3 | 23.99 | 20.29 | 27.69 | 23.97 | 20.94 | 27.01 | 23.98 | 20.62 | 27.35 |
| Salomon | 30.33 | 27.76 | 32.89 | 32.59 | 32.02 | 33.16 | 31.46 | 29.89 | 33.03 |
| Astley | 60.49 | 51.63 | 69.34 | 57.96 | 51.36 | 64.56 | 59.22 | 51.50 | 66.95 |

Research Agency, Spain MCIN/AEI/10.13039/ 501100011033 and the European Union "NextGenerationEU"/PRTR.

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
