# OpenReview forum: "Consistent Comparison of Symptom-based Methods for COVID-19 Infection Detection (Extended Abstract)"
_KDD.org/2023/Workshop/epiDAMIK — KDD 2023 Workshop epiDAMIK_

### Official Review · Reviewer_MW4n · 2023-06-27
**Consistent Comparison of Symptom-based Methods for COVID-19 Infection Detection- Review**

**Rating:** 4
**Confidence:** 3

**Review:**

In this work, the authors perform a consistent comparison of the different COVID-19 active case detection methods from the dataset constructed from the UMD-CTIS survey. The authors primarily implemented 3 broad types of detections, namely rule-based methods, logistic regression techniques and tree-based machine learning methods. F-1 score is used as the evaluation metric and the experiments were performed on the data from Brazil, Canada, Israel, Japan, Turkey and South Africa for the years 2020 and 2021.
Some of the comments for this work are as follows:
+ The work divides the data only in terms of years (2020 & 2021). However, a more important experiement would be to evaluate the models on the following scenarios:
     1. Beginning of COVID where information about the disease was not well known vs the time when we got a lot of info about COVID.
     2. Vaccines available vs not available.
     3. Different COVID variants (alpha, beta, etc).
     4. Model performance based on different mobility restrictions taking place (lockdowns, restricted international travel, etc)
+ AUC-ROC can also be considered to be an important evaluation metric to compare the performance of the different models.

---

### Official Review · Reviewer_rFQF · 2023-06-29
**This paper uses the UMD-CTIS dataset to evaluate the existing symptom-based detection methods.**

**Rating:** 3
**Confidence:** 5

**Review:**

This paper uses the UMD-CTIS dataset to evaluate the existing symptom-based detection methods.

Goodness:
1. The study covers many detection methods (10+) from three categories (rule-based methods, logistic regression-based methods, and tree-based methods), which gives an overview of the existing symptom-based detection methods.
2. The evaluation in both the 2020 period and 2021 period allows us to explore the influence of vaccines in symptom detection, which is especially useful since vaccines may make COVID-infected patients show fewer symptoms, which influences the detection method performance.

Weakness:
1. The result section only includes the table explanation and lists the performance of different methods, while a more detailed explanation of why some methods are better and the takeaways are missing.
2. The evaluation metric now is only the F1 score. More metrics are useful to better evaluate the difference between each method. Besides, in such detection problems, a high recall is usually more important than precision. More discussions can focus on this point.

---

### Official Review · Reviewer_oszk · 2023-06-30
**Consistent Comparison of Symptom-based Methods for COVID-19 Infection Detection**

**Rating:** 3
**Confidence:** 1

**Review:**

This paper compares the accuracy of many methods that detect COVID-19-positive cases. Most of these methods either propose simple rules or build machine learning models that determine a COVID-19-positive case based on certain individual attributes.

It is not entirely clear but I believe the authors train the ML based models on the same dataset (UMD-CTIS Survey) by splitting it randomly and using 80% for training and 20% for evaluating the performance. I do not understand exactly why this particular method of comparison is desirable.